# Knowledge of female genital schistosomiasis and urinary schistosomiasis among final-year midwifery students in the Volta Region of Ghana

**Wisdom Klutse Azanu**[1]☯, **Joseph Osarfo**[2]☯*, **Gideon Appiah**[3], **Yvonne Sefadzi Godonu**[4], **Gifty Dufie Ampofo**[2], **Verner Orish**[5], **Michael Amoh**[1], **Evans Kofi Agbeno**[6], **Emmanuel Senanu Komla Morhe**[1], **Margaret Gyapong**[7]

1 Department of Obstetrics and Gynaecology, School of Medicine, University of Health and Allied Sciences, Ho, Volta Region, Ghana, 2 Department of Community Health, School of Medicine, University of Health and Allied Sciences, Ho, Volta Region, Ghana, 3 Physician Assistantship Training Programme, School of Medicine, University of Health and Allied Sciences, Ho, Volta Region, Ghana, 4 Nursing and Midwifery Training College, Hohoe, Volta Region, Ghana, 5 Department of Microbiology and Immunology, School of Medicine, University of Health and Allied Sciences, Ho, Volta Region, Ghana, 6 Department of Obstetrics and Gynaecology, School of Medical Sciences, University of Cape Coast, Cape Coast, Central Region, Ghana, 7 Institute of Health Research, University of Health and Allied Sciences, Ho, Volta Region, Ghana

☯ These authors contributed equally to this work.
* josarfo@uhas.edu.gh

**Data Availability Statement:** All relevant data are within the manuscript and its Supporting Information files.

## Abstract

### Background

Female genital schistosomiasis (FGS) is a gynaecological complication of urinary schistosomiasis (US) with an estimated burden of 20–120 million cases in endemic areas. A neglected sexual and reproductive health disease in sub-Saharan Africa, FGS increases susceptibility to sexually transmitted infections including cervical cancer and infertility among other morbidities. However, there appears to be limited FGS knowledge among practicing and pre-service health providers with implications for control. We assessed FGS awareness among final-year midwifery students who would soon be delivering primary maternal and reproductive health care.

### Methods

A cross-sectional study was conducted among 193 randomly selected final-year students from all three midwifery training institutions in the Volta region of Ghana in August/September, 2022. Data on participants' demographics and knowledge of the transmission, signs and symptoms, complications, treatment and prevention of both FGS and US were collected using structured questionnaires. Summary statistics were presented as frequencies, proportions and percentages.

**Funding:** The author(s) received no specific funding for this work.

**Competing interests:** The authors have declared that no competing interests exist.

## Results

Only 23.3% (44/189) of participants had heard about FGS compared to 64% (123/192) for US. Of the former, 42 (95%), 40 (91%) and 36 (81.8%) respectively identified genital itching/burning sensation, bloody vaginal discharge and pelvic pain/pain during intercourse as part of the symptoms of FGS. Less than a third (13/44) and about half (25/44) of those who indicated hearing about FGS knew it can be a risk for ectopic pregnancies and infertility respectively. Majority of these participants, 40 (91%), wrongly selected antibiotics as treatment for FGS while 9 indicated it is prevented by sleeping in insecticide-treated nets.

## Conclusion

Awareness of FGS was limited among the study participants. The high prevalence of knowledge of some FGS symptoms related to the genitalia needs cautious interpretation. Health care training institutions must make deliberate efforts to highlight FGS in the training of midwives as the condition has diagnostic and management implications for some sexual and reproductive health conditions.

## Introduction

Female genital schistosomiasis (FGS) is the gynaecological manifestation of urinary schistosomiasis which is originally caused by *Schistosoma haematobium* infection. Whereas urinary schistosomiasis affects over 200 million people in the world [1] with 95% of the global burden occurring in sub-Saharan Africa, there are about 20 to 120 million cases of FGS in this endemic region [2]. These figures may even be underestimates as FGS is grossly underreported probably due to poor diagnostic capacity resulting from inadequate knowledge or a lack of awareness among health workers in endemic regions [2]. Female genital schistosomiasis is characterized by the deposition of schistosoma eggs in the female genital tract [3] and this process may underlie the clinical symptoms of vaginal bleeding, pain during sexual intercourse, genital sores and nodules in the vulva among others [4]. Female genital schistosomiasis has been described as one of the most neglected sexual and reproductive health diseases and it is rarely considered among females with urinary schistosomiasis in endemic areas [2]. In a bid to increase its awareness, the World Health organization (WHO), in 2009, recommended that urinary schistosomiasis be officially referred to as urogenital schistosomiasis [5]. Female genital schistosomiasis increases susceptibility to sexually transmitted infections (STIs) [6]. Women and girls with FGS have a three times higher risk of acquiring human immunodeficiency virus (HIV) and twice the risk for human papilloma virus HPV) infections [7]. Female genital schistosomiasis can be misdiagnosed as an STI and this can lead to depression and social stigma in girls and women because of negative societal perception of having an STI. Misdiagnosis also means wrong treatment and hence FGS symptoms linger on and underlie increased risk of HIV infection, gynaecological complications like infertility, ectopic pregnancy, abortions and increasing the risk of cervical cancer [2, 8].

For urinary/ intestinal schistosomiasis, an awareness of 75%-95% among the general population has been reported in sub-Saharan Africa in a systematic review but adequate knowledge on signs and symptoms, transmission and disease control is low [9] and is fraught with misconceptions such as assertions that transmission occurs by having unprotected sex, ingesting contaminated food and water and witchcraft [10–12]. This is contrary to expectations

considering that the region is highly endemic. No published study was found on the knowledge of urinary schistosomiasis among midwifery students but more than half of female medical, pharmacy and other biomedical science students were not aware of urinary schistosomiasis in Nigeria [13]. Even among clinicians, only 46.7% had good knowledge of treatment of urinary schistosomiasis while 56.7% had fair knowledge of its prevention [14]. A much higher prevalence of knowledge of urinary schistosomiasis, 96.9%, has been reported among health care workers though [13]. Urinary schistosomiasis remains prevalent all over Ghana with increased presence in surrounding areas of the Volta Lake [15, 16]. A prevalence of about 47% was reported among adults living in the Volta Basin of Ghana about a decade ago [17]. More recently, a prevalence of about 7.6% has been observed among female adolescents, 11–15 years old, in the Oti Region which was formerly part of the Volta Region [18]. Many Ghanaian studies on urinary schistosomiasis have hinged on its distribution, determinants, diagnosis and control to the neglect of the complication of FGS [15, 16] and this may explain why very little is known of the burden of this specific complication in the country.

Over the past decade, there have been increased efforts at increasing the awareness and knowledge of FGS in terms of diagnostic research [19–22] and pragmatic policy guides and information [7, 23] but there is still evidence that only a very small fraction of those with suspected FGS have been examined clinically [7, 20, 23]. This gap could be due to the fact that health care workers at the primary health care centers in most endemic communities are largely unaware of the existence or diagnosis of this important debilitating gender-specific manifestation of urogenital schistosomiasis [15, 24]. In a qualitative study among health workers in Tanzania [25], most participants reported that they were hearing about FGS for the first time in the study. In addition, these participants were unaware that schistosomiasis could affect the male and female reproductive systems. They also had misconceptions regarding the transmission of FGS, particularly stating that it can be transmitted through sexual intercourse [25].

A Nigerian study profiling the knowledge of female medical/para-medical students and health care professionals on female genital schistosomiasis reported that over half of the students were not aware of the condition [13]. Similarly, the health care workers had a relatively low knowledge about FGS (61.9%) [13]. Kukula et al. [15] assessed key gaps in knowledge and understanding of female genital schistosomiasis among community members and local health workers in Ghana and reported that most frontline health workers lacked knowledge on the causes, prevention, transmission, symptoms and complications of FGS. Only one gynecologist and a midwife, who worked in a cervical screening clinic, expressed a very good understanding of FGS in this Ghanaian study [15]. Clearly, a lack of or inadequate knowledge / training among health care providers regarding FGS will likely fuel misdiagnosis and become a challenge for disease control.

Final-year midwifery students in Ghana constitute a pre-service group who, upon completion of their programme of study in their various training institutions, will add on to the workforce at the various hospitals, clinics /health centers, and community-based health planning and services (CHPS) compounds to deliver primary health care (PHC). The CHPS compounds at the community level and health centers at the sub-district level form the basic levels of the health system in Ghana [26]. In many communities, the midwives serve as the first point of call either through the CHPS compounds or the clinics. If this cadre of health workers are aware of FGS, they can at least suspect it as a diagnosis and make a timely referral for the patients to be diagnosed on time and thus preventing long term complications. There is no data on the knowledge of these pre-service midwives in Ghana about FGS. Largely, the majority of studies reporting on knowledge or awareness of FGS have centered on practicing health workers with rather limited involvement of student health workers in training.

It is therefore important to assess the knowledge of health care providers, including those still in training, regarding FGS. The study sought to assess the knowledge of female genital schistosomiasis and urinary schistosomiasis among final year midwifery students in the Volta region of Ghana. The Volta region was chosen because communities along the Volta Lake have been known to be endemic for urinary schistosomiasis [15, 27] and also for ease of accessibility to the students as all but one of the investigators are based in the region. With FGS being a complication of urinary schistosomiasis, assessing the awareness of urinary schistosomiasis adds more information and offers a platform of comparison of knowledge base. It also gives an idea of how much information is needed to be given to this group of midwives about FGS. The findings from the study will help in determining and planning further training for these groups of health workers even before they are posted especially to schistosoma endemic areas. It will also serve as a feedback tool to the training institution and will aid decision processes on a potential need to review the curriculum currently being used in the training of midwives in Ghana.

## Methodology

### Study design

This was a cross-sectional study conducted in all three midwifery training institutions in the Volta region of Ghana.

### Description of study sites

The survey was conducted at 3 different sites; the Hohoe Nurses and Midwifery Training College (NMTC), the Keta NMTC and the School of Nursing and Midwifery at the University of Health and Allied Sciences in Ho.

Hohoe is situated in the middle of the Volta region. Out-patient department (OPD) records from the Hohoe hospital showed that out of the 6cases of urinary schistosomiasis diagnosed in 2021, 4 were females. In 2022, there was only 1 female out of 15 cases of urinary schistosomiasis (Biostatistics Unit, Hohoe Hospital, 2022).

Ho is the capital of the Volta Region and records from the district health information management system (DHIMS 2), Ghana's electronic database of morbidity, mortality and health services utilization, showed that there were 7 and 28 cases of urinary schistosomiasis diagnosed in 2021 and 2022 respectively in this municipal area. Of these, there was 1 female in 2021 and 23 in 2022.

The Keta municipality houses the Keta NMTC. About 30% of the total surface area of the Keta municipal area is covered by water bodies, the largest of which is the Keta lagoon. The inhabitants are mainly farmers and fishermen. Data from the Keta municipal hospital shows that there were 9 females out of 20 cases in 2021 and 2 females out of 11 cases of urinary schistosomiasis in 2022 (Biostatistics Unit, Keta Municipal Hospital, 2022).

### Study population

The study population comprised all final-year midwifery students in the Volta region.

### Sample size estimation

The sample size for the study was calculated using Yamane's Formula;

$n = N/(1+N(e)2$ where **n** is sample size, **N** is population size and **e** is the desired margin of error.

The population size, being the total number of final-year midwifery students in all three sites, was 312. Using a margin of error of 5%, a sample size of 175 was obtained. Adjusting for a 10% non-response rate gave a final figure of 193. Using a proportion to size for the final year class of each institution gave 71, 73 and 49 respondents to be recruited from Hohoe-NMTC, Keta-NMTC, and UHAS respectively.

## Study procedures and data collection

The study was conducted from 22$^{nd}$ August, 2022 to 12$^{th}$ September, 2022. All final-year midwifery students in the Volta region were eligible for inclusion in the study. Those who were below 18 years of age were excluded from the study as they were minors and required consent from their parents / guardians. This would have been operationally difficult to obtain. Students who were not available on the day of the survey were also excluded from the study. The study team moved to one institution, collected the data within a day and moved to the next institution the following week. At each study site, the team met with the final-year midwifery students in a class and the study objectives were explained. Those who agreed to partake in the study remained in the classroom while the other students moved out. Study participants were selected from those that chose to partake in the study using simple random selection methods. Pieces of papers with 'yes' and 'no' written on them were folded for subsequent selection with non-replacement. The study team ensured that the number of 'yes' was equal to the determined sample size for each institution. Those who picked 'yes' stayed behind to complete the self-administered questionnaire while those who picked 'no' moved out of the class. The team also ensured that none of the participants made any consultation from their phones, materials or friends in filling the questionnaire.

The structured questionnaire used for data collection (see S1 Questionnaire) was developed using the information contained in a teaching manual for urinary schistosomiasis and FGS [28] as a guide. Data was collected on knowledge and awareness of urinary schistosomiasis and female genital schistosomiasis. Specifically, the questionnaire assessed mainly participants' demographics, awareness, knowledge of signs and symptoms, complications, transmission, treatment and the prevention of urinary schistosomiasis and FGS. The questionnaire was divided into three main parts. The first part collected data on the background characteristics of participants which included age, and institution attended and whether they had ever heard of urinary and female genital schistosomiasis. The second part assessed the awareness, knowledge of signs and symptoms, complications, transmission, treatment and the prevention of urinary schistosomiasis using close-ended questions. The third part assessed awareness, knowledge of signs and symptoms, complications, transmission, treatment and the prevention of female genital schistosomiasis also using close-ended questions. The questionnaire was pretested among 10 final year general nursing students of the Ho Nursing Training College who were having their clinical rotation at the Ho Teaching Hospital and relevant changes made to the structure and content of the questionnaire before it was used in the main study.

## Data management and analysis

The filled questionnaires were checked for completeness and accuracy. Data was entered in Microsoft Excel and exported to Stata version 13 (Stata Corp, TX, USA) for analysis. Descriptive statistics were done and presented as frequencies, proportions and percentages in tables. Chi-square test was used to assess for association between the independent variables 'school attended' and 'age group' and the dependent variables 'ever heard of urinary schistosomiasis' and 'ever heard of FGS'. An association was said to be statistically significant if $p < 0.05$.

### Ethical consideration

Ethical clearance for the study was granted by the Research Ethics Committee of the University of Health and Allied Sciences (Protocol identification number; UHAS-REC A.11 (88) 21–22). Permission was also obtained from the principals of the two midwifery training schools and the head of the midwifery department of the University of Health and Allied Sciences. Written informed consent was obtained from each participant following explanation and clarification of the purpose of this study and it was affirmed that the study posed no risks. Study participants were anonymized by assigning them study codes to ensure confidentiality. Participants were informed that they were free to withdraw from the study at any point without penalty. There was no compensation package offered to the participants.

## Results

### Background characteristics of participants

All planned 193 final-year students took part in the survey. Hohoe and Keta NMTC had similar numbers of participants in the study; 71 (36.8%) and 73 (37.8%). Close to half (48.7%) were in the age group 23–27 years while 31 (16.1%) were in the youngest age group, 18–22 years. One hundred and twenty-three (64.1%) participants indicated they had heard about urinary schistosomiasis. Comparatively, just about a quarter (23.3%) reported ever hearing of female genital schistosomiasis (see Table 1).

In the Chi-square analysis, Keta NMTC had the highest proportion of participants who had heard about urinary schistosomiasis followed by UHAS and Hohoe respectively and this was statistically significant (47.2% vs 30.0% vs 22.8%; p = 0.002) (See Table 3). With FGS, there was no difference in the proportions of participants from the different schools who had heard about it. Similarly, there was a statistically significant association between the participants' age categories and whether they had ever heard of urinary schistosomiasis. Almost 50% of participants who indicated they had heard of urinary schistosomiasis were in the age group 23–27 years. However, there was no such age relation with FGS (See Table 2).

Table 1. Background characteristics of participants.

| Variable | Frequency | percentage % |
|---|---:|---:|
| **Age (years) (N = 193)** | | |
| 18–22 | 31 | 16.1 |
| 23–27 | 94 | 48.7 |
| 28–32 | 29 | 15.0 |
| ≥33 | 39 | 20.2 |
| **School (N = 193)** | | |
| Hohoe-NMTC | 71 | 36.8 |
| Keta-NMTC | 73 | 37.8 |
| UHAS | 49 | 25.4 |
| **Ever heard of Urinary Schistosomiasis (N = 192)** | | |
| No | 69 | 35.9 |
| Yes | 123 | 64.1 |
| **Ever heard of Female Genital Schistosomiasis (N = 189)** | | |
| No | 145 | 76.7 |
| Yes | 44 | 23.3 |

**Table 2.** Association between the independent variables 'Age group' and 'School attended' and the dependent variables 'Ever heard of urinary schistosomiasis' and 'Ever heard of female genital schistosomiasis'.

| Variable | Ever heard of Urinary Schistosomiasis (N = 192) | | | Ever heard of Female Genital Schistosomiasis (N = 189) | | |
|---|---|---|---|---|---|---|
| | **Yes n (%)** | **No n (%)** | **p-value** | **Yes n (%)** | **No n (%)** | **p-value** |
| **Age group (years)** | | | 0.019 | | | *0.253 |
| 18–22 | 15 (12.2) | 15 (21.8) | | 6 (13.6) | 25 (17.2) | |
| 23–27 | 61 (49.6) | 33 (47.8) | | 25 (56.8) | 67 (46.2) | |
| 28–32 | 15 (12.2) | 14 (20.3) | | 3 (6.8) | 26 (18.0) | |
| ≥ 33 | 32 (26.0) | 7 (10.1) | | 10 (22.8) | 27 (18.6) | |
| **School** | | | 0.002 | | | 0.458 |
| Hohoe | 37 (30.0) | 33 (47.8) | | 20 (45.5) | 51 (35.2) | |
| Keta | 58 (47.2) | 15 (21.8) | | 14 (31.8) | 57 (39.3) | |
| UHAS | 28 (22.8) | 21 (30.4) | | 10 (22.7) | 37 (25.5) | |

*Fisher's exact p-value reported

## Participants' source of information about urinary and female genital schistosomiasis

Of the 44 who reported hearing about FGS, 26 (59.1%) indicated they had heard about it from school/workshops while 14 (31.8%) named hospitals or the mass media as their sources (See Table 3). About 4.6% (2) did not remember where they had heard about the condition. Similarly, of the 122 students who reported hearing of urinary schistosomiasis, 67(55%) indicated they had heard about it from school/workshops while 22(18%) could not remember where they had heard about it from.

## Knowledge of urinary schistosomiasis

The results presented here relate to the 122 participants who indicated they had ever heard of urinary schistosomiasis. Approximately 93% (114) were correctly able to tell that blood in urine (haematuria) was a symptom of urinary schistosomiasis (see Table 4). About 60% (70), 30% (36) and 23% (28) respectively selected fever, difficulty in walking and inguinal hernia as part of the signs and symptoms of US though these were not correct.

Hindering growth of children and their cognitive function and bladder tumour are true complications of US but they were each selected by less than 50% of participants. Seventy-nine (64.8%) participants correctly selected occurrence of FGS as a fall out of US (see Table 4).

## Knowledge of female genital schistosomiasis

The results presented in this sub-section pertains solely to the 44 participants who indicated that they had heard of FGS before. Of these 44, 41(92.3%) were part of the 122 respondents

**Table 3.** Source of information on urinary and female genital schistosomiasis among participants who reported hearing about them.

| Source | Urinary Schistosomiasis (N = 122) n(%) | Female Genital Schistosomiasis (N = 44) n(%) |
|---|---|---|
| Family / Friends | 2 (1.6) | 2 (4.6) |
| Health Facility | 25 (20.5) | 8 (18.2) |
| I do not remember | 22 (18.1) | 2 (4.6) |
| Mass media | 6 (4.9) | 6 (13.6) |
| School / Workshop | 67 (54.9) | 26 (59.0) |

**Table 4. Knowledge of urinary schistosomiasis from participants\* who reported they had heard of it before.**

| Knowledge components assessed | Participants' responses (N = 122) | | |
|---|---|---|---|
| **Signs and symptoms of Urinary Schistosomiasis** | Yes n(%) | No n(%) | Don't know n(%) |
| Blood in urine | 114 (92.7) | 4 (3.3) | 5 (4.0) |
| Diarrhoea | 18 (4.4) | 74 (59.2) | 31 (26.4) |
| Fever | 70 (56.9) | 28 (22.8) | 25 (20.3) |
| Burning urination | 104 (84.5) | 7 (5.7) | 12 (9.8) |
| Fatigue | 42 (34.2) | 42 (34.2) | 39 (31.6) |
| Loss of appetite | 32 (26.0) | 49 (39.8) | 42 (34.2) |
| Itching | 82 (66.7) | 20 (16.3) | 21 (17.0) |
| Difficulty in walking | 36 (29.8) | 53 (43.8) | 32 (26.4) |
| Inguinal Hernia | 28 (22.8) | 55 (44.7) | 40 (32.5) |
| **Complications of Urinary Schistosomiasis** | | | |
| It stops children from growing well | 45 (36.9) | 44 (36.1) | 33 (27.0) |
| Heart attack | 24 (19.2) | 64 (51.2) | 37 (29.6) |
| HIV | 14 (11.5) | 70 (57.4) | 38 (31.2) |
| It may lead to female genital schistosomiasis | 79 (64.8) | 14 (11.4) | 29 (23.8) |
| Bladder tumour | 61 (49.6) | 30 (24.4) | 32 (26.0) |
| Liver failure | 48 (39.0) | 39 (31.7) | 36 (29.3) |
| It stops children from learning well | 43 (35.0) | 44 (35.8) | 36 (29.2) |
| **Transmission of Urinary Schistosomiasis** | | | |
| Drinking contaminated water | 84 (68.9) | 20 (16.4) | 18 (14.7) |
| Eating unwashed food | 47 (38.2) | 46 (37.4) | 30 (24.4) |
| Walking barefoot in contaminated soil | 52 (42.6) | 38 (31.2) | 32 (26.4) |
| Mosquito bite | 8 (6.6) | 74 (61.7) | 38 (31.7) |
| Contact with contaminated water | 88 (71.5) | 13 (10.6) | 22 (17.9) |
| Insulting the gods | 5 (4.1) | 81 (65.9) | 37 (30.1) |
| **Treatment of Urinary Schistosomiasis** | | | |
| Antibiotics | 92 (74.2) | 5 (4.0) | 27 (21.8) |
| Herbs | 14 (11.4) | 58 (47.1) | 51 (41.5) |
| Praziquantel | 49 (39.8) | 35 (28.5) | 39 (31.7) |
| It goes on its own | 7 (5.7) | 67 (55.0) | 48 (39.3) |
| Antivirals | 24 (19.7) | 51 (41.8) | 47 (38.5) |
| Antifungal | 34 (27.4) | 41 (33.1) | 49 (39.5) |
| **Prevention of Urinary Schistosomiasis** | | | |
| Wearing condom before sexual intercourse | 61 (49.6) | 30 (24.4) | 32 (26.0) |
| Avoid urinating into water bodies | 76 (62.3) | 16 (13.1) | 30 (24.6) |
| Avoid swimming or playing in contaminated water | 85 (69.7) | 7 (5.7) | 30 (24.6) |
| Boil water from contaminated source before using | 87 (71.9) | 8 (6.6) | 26 (21.5) |
| Avoid defecating into water bodies | 76 (62.3) | 10 (8.2) | 36 (29.5) |
| Sleeping under insecticide treated nets | 19 (15.7) | 57 (47.1) | 45 (37.2) |
| Vaccination | 48 (39.3) | 33 (27.1) | 41 (33.6) |
| Using insecticides | 22 (18.0) | 54 (44.3) | 46 (37.7) |
| Mass treatment of communities with praziquantel | 59 (48.4) | 27 (22.1) | 36 (29.5) |

who had earlier indicated they had ever heard of urinary schistosomiasis. The remaining 3 reported they had heard of FGS even though they had never heard of urinary schistosomiasis before. The numbers/percentage of participants that selected 'yes', 'no' and 'I don't know' to the questions relating to knowledge of FGS are shown in Table 5. About 86.4% (38) of this

**Table 5. Knowledge of female genital schistosomiasis among the 44 participants who reported they had heard of FGS before.**

| Knowledge Component Assessed | Participant Responses | | |
|---|---|---|---|
| **Late detection or untreated FGS can lead to;** | Yes n(%) | No n(%) | I don't know n(%) |
| Fibroids | 8 (18.6) | 24 (55.8) | 11 (25.6) |
| Cervical Cancer | 27 (60.0) | 9 (20.0) | 8 (20.0) |
| Gonorrhoea | 20 (45.5) | 14 (31.8) | 10 (22.7) |
| Syphilis | 23 (52.3) | 12 (27.3) | 9 (20.4) |
| Candidiasis | 21 (47.7) | 14 (31.8) | 9 (20.5) |
| Ectopic Pregnancy | 13 (29.6) | 19 (43.2) | 12 (27.2) |
| Infertility | 25 (56.8) | 9 (20.5) | 10 (22.7) |
| Abortion | 17 (38.6) | 14 (31.8) | 13 (29.6) |
| Genital Ulcers | 28 (63.6) | 4 (9.1) | 12 (27.3) |
| **FGS is transmitted in the following ways;** | | | |
| playing with soil | 17 (38.6) | 15 (34.1) | 12 (27.3) |
| eating contaminated food | 14 (31.8) | 17 (38.6) | 13 (29.6) |
| drinking untreated water | 26 (59.0) | 9 (20.5) | 9 (20.5) |
| unprotected sexual intercourse | 24 (54.6) | 12 (27.3) | 8 (18.1) |
| swimming/bathing in infested water | 33 (75.0) | 4 (9.1) | 7 (15.9) |
| dirty hands | 14 (31.8) | 15 (34.1) | 15 (34.1) |
| physical contact with an infected person | 17 (38.6) | 14 (31.8) | 13 (29.6) |
| improper use of family planning methods | 5 (11.4) | 27 (61.4) | 12 (27.2) |
| punishment from the gods | 2 (4.6) | 30 (68.2) | 12 (27.2) |
| **The signs and symptoms of FGS include;** | | | |
| vaginal discharge | 34 (77.3) | 6 (13.6) | 4 (9.1) |
| bloody vaginal discharge | 40 (90.9) | 4 (9.1) | 0 (0.0) |
| bleeding after intercourse or spotting | 29 (65.9) | 9 (20.5) | 6 (13.6) |
| genital itching or burning sensation | 42 (95.4) | 1 (2.3) | 1 (2.3) |
| pelvic pain or pain during/after intercourse | 36 (81.8) | 3 (6.8) | 5 (11.4) |
| **FGS is treated using;** | | | |
| antibiotics | 40 (90.9) | 1 (2.3) | 3 (6.8) |
| praziquantel | 23 (52.3) | 11 (25.0) | 10 (22.7) |
| herbal medications | 6 (13.6) | 26 (59.1) | 12 (27.3) |
| antivirals | 10 (22.7) | 21 (47.7) | 13 (29.6) |
| contraceptives | 8 (18.1) | 23 (52.3) | 13 (29.6) |
| antifungals | 20 (45.4) | 12 (27.3) | 12 (27.3) |
| **FGS can be prevented in the following way;** | | | |
| wearing condom before sex | 27 (61.4) | 11 (25.0) | 6 (13.6) |
| avoid urinating into water bodies | 33 (75.0) | 5 (11.4) | 6 (13.6) |
| avoid swimming or playing in infested water | 40 (90.9) | 1 (2.3) | 3 (6.8) |
| avoid defecating into water bodies | 34 (77.3) | 3 (6.8) | 7 (15.9) |
| boil water from contaminated source before using | 36 (81.8) | 4 (9.1) | 4 (9.1) |
| sleeping under insecticide treated net | 9 (20.4) | 24 (54.6) | 11 (25.0) |
| vaccination | 22 (50.0) | 13 (29.6) | 9 (20.4) |
| mass treatment of communities with praziquantel | 27 (61.4) | 9 (20.4) | 8 (18.1) |

sub-group correctly identified ectopic pregnancy and infertility as complications of FGS while 24 (54.6%) incorrectly reckoned FGS could be transmitted through unprotected sexual intercourse. About 91% (40) of these particular participants thought FGS can be treated can be treated with antibiotics.

## Discussion

To help fill existing knowledge gaps, this study assessed the knowledge of final-year midwifery students in the Volta region of Ghana with respect to the signs and symptoms of FGS, its complications, treatment and prevention. Approximately 82% and 77% of respondents who reported hearing about FGS previously correctly selected pain during or after sex and vaginal discharge respectively as presenting symptoms. A little over 60% of respondents who reported hearing about FGS previously indicated protected sex with condoms as a preventive measure. Similarly, 50% of this population also indicated the use of vaccines as a preventive measure against FGS. While only a little over half of respondents who reported hearing about FGS correctly indicated Praziquantel as the drug for treatment, about 91% indicated antibiotics as the treatment of choice. Among those who reported hearing about urinary schistosomiasis before, about 93% correctly chose 'blood in urine' as a symptom while 68% wrongly selected inguinal hernia as a symptom or indicated they did not know if it was or not. Nearly 70% of these specific participants indicated that urinary schistosomiasis was transmitted by drinking contaminated water.

Although a number of studies have been conducted in Ghana on knowledge, attitudes and practices in relation to urinary schistosomiasis [2, 27, 29], not much has been done about FGS among pre-service health workers. The current study is the first to describe the awareness of FGS among pre-service midwives in Ghana. Although 122 of the respondents indicated they had heard about urinary schistosomiasis, only just about a quarter (23.3%) reported ever hearing of FGS. This reflects the general trend where more people are aware of urinary schistosomiasis than FGS [13, 15, 29]. This trend could also arise from a lot of awareness creation and community efforts towards elimination of urinary schistosomiasis compared to FGS which, until recently, was not talked about much or at all [24]. Relatively low FGS awareness among community members, health workers and medical/para-medical students has been reported in Ghana, Tanzania and Nigeria [15, 13, 25]. In the Nigerian study [13], nearly 42% of the participants were aware of FGS and this is higher than the 23.3% observed in the current study. The difference could be due to the fact that the study population in the Nigerian study included medical and paramedical students so a wider scope of recruitment means that some of the students may have heard of it from the classroom based on their curricula. In the current study, only midwifery students were studied so there could be limited level of awareness.

Majority of those who were aware of FGS (65.4%) in the study by Aribodor et al. [13] first heard of it from the school environment and this compares favourably with the present study where 59.1% of the respondents also heard of it from school/workshops. "School/workshop" was the commonest source of information concerning urinary schistosomiasis and FGS in the present study. Urinary schistosomiasis has been a part of the science curriculum in senior high schools and also health training institutions in Ghana. This makes it likely that these pre-service midwives would have heard about urinary schistosomiasis aside whatever exposure they may have had in the clinical setting. In comparison, FGS is so far not part of the curriculum in their training. It is probable then that the knowledge of FGS was more from extracurricular workshops organized for these midwives. For instance, it is possible that their tutors may have been involved in training workshops about schistosomiasis and FGS organized by the Volta River Authority (VRA) and may have casually handed down such information during lectures on related topics such as anaemia in pregnancy. In addition to the Neglected Tropical Diseases Control Programme in Ghana, the VRA is also involved in schistosomiasis control along the Volta Lake. Nevertheless, here in Ghana, the named sources may have to be taken with caution since FGS is still a relatively new 'discovery' that is now being brought to the attention of even clinicians. This is reinforced by findings in a Ghanaian study [15] where only one doctor and

one midwife working in a cervical cancer clinic had thorough knowledge of FGS. It is reasonable to assume that these health workers in reference [15] had been given specialized training considering the setting in which they worked. It is unlikely that health education on FGS has featured to any appreciable extent in the mass media and the respondents probably selected those options simply to complete filling the questionnaire.

Keta NMTC had the highest proportion of participants who had heard about urinary schistosomiasis followed by UHAS and Hohoe NMTC respectively. This could be because Keta has a lot more water bodies than the other places and possibly a higher burden of urinary schistosomiasis which may underpin relatively more awareness among these students as their training also involves the hospital and community setting. The mass drug administration for prevention of *Schistosoma* infection in that area may also have created more awareness. Also in Keta NMTC, most of the students live within the community rather than in secluded hostels and so may be more aware of or involved in community activities pertaining to urinary schistosomiasis than in UHAS and Hohoe. This relation between the school attended and whether the respondents had ever heard of urinary schistosomiasis was not observed in the case of FGS. This may arise from the general deficiency in knowledge of FGS.

The highest proportion, about half, of those who had heard about urinary schistosomiasis were in the age group 23–27 years and the next highest proportion was in the ≥33 year group. While the observation for the age category 23–27 years may arise from the fact that it has the highest population of participants, the observation in the group ≥33 years could be plausibly linked to the common practice where older students may have worked in health facilities for some period before enrolling in the midwifery school and have therefore had myriad encounters with urinary schistosomiasis patients compared to the students in the younger age groups. The latter likely came into the training school straight from senior high school and may have had little or no exposure to the work environment to enable them possess adequate knowledge about urinary schistosomiasis.

Of the 44 participants that have ever heard of FGS, thirty-three (75%) correctly selected swimming/bathing in contaminated water as a mechanism of transmission for FGS. This relatively high score compared to the other statements in this segment could probably be due to the fact that the respondents were able to link urinary schistosomiasis to FGS thereby assuming a similar causal/transmission mechanism. It is noteworthy that even though wrong options including 'eating contaminated food', 'playing with soil', 'unprotected sexual intercourse' and 'improper use of family planning methods' were selected as correct by some participants, comparable numbers of participants and even more, in some cases, also correctly identified them as wrong. Some wrong choices selected as correct raise questions as to whether the participants had truly heard of FGS or to what extent their awareness/knowledge of the condition was. For instance, drinking untreated/contaminated water was selected by about 60% of the participants as a method of transmitting FGS.

Regarding potential complications, only about 30% and 57% of the participants, respectively, were able to identify that FGS could lead to ectopic pregnancy events and infertility. All other options in this category were wrong but still got selected by appreciable numbers of participants. Almost half of those who reported hearing of FGS chose gonorrhoea and syphilis as a complication. This buttresses the observation and the wrong impressions noted in several other studies where health workers do not even consider FGS as a differential diagnosis when patients symptomatic for FGS present to the clinic [13, 15, 25, 27].

There is literature linking FGS to cervical cancer [30–33]. However, since these students typically don't read published research articles, such literature are unlikely to be the reason why 60% of those who were aware of FGS chose cervical cancer as a complication of FGS. They probably might have linked the 'genital' in FGS to the occurrence of cervical cancer. The

egg deposition in the vulva and vaginal tissue results in the sandy patches and rubbery papules which cause the vaginal wall to become brittle and is associated with contact bleeding [3, 34], There is no classical ulceration associated with FGS.

All options presented under signs and symptoms of FGS were correct and were appropriately selected by the majority of the participants in question. Almost all participants selected 'genital itching and burning sensation' while about 91% selected 'bloody vaginal discharge' and 81.8% selected 'pelvic pain or pain during/after intercourse' as correct responses. This level of good knowledge of the signs and symptoms contrasts with the findings of a qualitative study done in Shai-Osudoku in the Greater Accra region of Ghana among community members and local health workers [15] where the majority of the participants had poor knowledge of the signs and symptoms of FGS. The apparently better knowledge of FGS signs and symptoms demonstrated in the current study, about 91% for bloody vaginal discharge and 82% for pain during or after sexual intercourse among others, could be because the respondents had associated female genital schistosomiasis with pelvic inflammation and/or sexually transmitted diseases such as gonorrhea and subsequently inferred its signs and symptoms.

Again, judging from the high proportions of participants who selected correct responses regarding the signs and symptoms of FGS and comparing those to the selection of wrong options in other sections of the FGS knowledge assessment tool, the conclusion that the participants made a link between the 'genital' in FGS and the correct choices of vaginal discharge, bloody vaginal discharge, genital itching, pain at/during sex, etc seems to be a reasonable one. It is also possible that it is for this reason that 54.6% of respondents may have selected unprotected sexual intercourse as a mechanism of transmission for FGS.

The right option for treatment of FGS, Praziquantel, was chosen by 52.3% of participants who indicated they had heard of FGS while 91% and 45% wrongly selected antibiotics and antifungals respectively as correct responses. This implies that majority of them did not know that treatment was with Praziquantel. A similar lack of knowledge was also exhibited by health workers in a study in Uganda [10]. The huge proportion selecting antibiotics as the treatment for FGS is worrying and can be linked to participants seeing the condition as a bacterial infection or simply as an 'infection' which calls for use of antibiotics as is commonly perceived. The Schistosoma parasite is not a bacterium and thus does not require use of antibiotics.

Considering prevention of FGS, majority of participants correctly selected 'avoid urinating into water bodies' (75%), avoid swimming/playing in contaminated water' (90.9%), 'avoid defecating into water bodies' (77.3%) and 'boil water from contaminated source before using' (81.8%). Mass drug administration with Praziquantel in endemic areas is also a way of preventing urinary schistosomiasis from progressing to FGS in females and this was selected by 61.4% of the participants. The discrepancy between the proportions of participants who indicated awareness of mass drug administration and those who selected the other preventive methods highlight a need for improved public education on FGS and its prevention. FGS prevention with Praziquantel has been considered important and on the same scale as HPV vaccination against cervical cancer in efforts to safeguard the reproductive and sexual health of women in endemic areas [35]. Nine participants wrongly selected 'sleeping under insecticide-treated nets' as a way to prevent FGS. While this may not appear important on account of the low numbers, it is worrisome that final-year midwifery students would select such a glaring wrong response.

About 64% (123) of participants in the present study indicated they had heard about urinary schistosomiasis. This is higher than the findings in a similar study in Nigeria where just about half of the students 54.2% affirmed they have heard of urinary schistosomiasis [13]. The relatively higher awareness in this current study could be due to the fact that some of the midwifery students had done some community health nursing before enrolling in the midwifery

school so they could have had more exposure to the mass drug administration which is practised in Ghana. Approximately 93% of the respondents who were aware of urinary schistosomiasis were correctly able to tell that blood in urine (haematuria) was a symptom of US. This finding is similar to findings of several other works in Tanzania, Zimbabwe and Ghana where majority of the respondents also identified blood in urine as a symptom of urinary schistosomiasis [6, 29, 36].

The respondents exhibited poor knowledge in terms of knowledge of the signs and symptoms of urinary schistosomiasis. About 60%, 30% and 23% respectively selected fever, difficulty in walking and inguinal hernia as part of the signs and symptoms of US but these were not correct. These findings were similar to findings in previous studies including a systematic review [9, 25].

Hindering growth of children and their cognitive function and bladder tumour are true complications of urinary schistosomiasis [37] but they were each selected by less than 50% of participants. This suggests that they had relatively poor knowledge regarding the complications of even urinary schistosomiasis. About 65% of participants who indicated they had heard of urinary schistosomiasis correctly selected occurrence of FGS as a complication. While this may be a clever guess derived from probably linking the two conditions on the commonality of 'schistosomiasis, it also indicates a lack of comprehensive knowledge of urinary schistosomiasis and its complications among the participants. One may contest the necessity in requiring midwifery students to have a more-than-average knowledge of urinary schistosomiasis on grounds that it may not be directly linked to delivery of maternal and newborn care. However, we posit that such knowledge is important for comprehensive reproductive and sexual health services. While all Schistosome species may cause some degree of hepatobiliary disease, *S. hematobium* is less likely to do so, and affects primarily the urinary tract. The main species that cause hepatobiliary disease are *S. mansoni* and *S. japonicum*. Liver damage is therefore not generally considered as a complication of urinary schistosomiasis [38].

Majority of respondents who indicated they had heard of urinary schistosomiasis wrongly selected antibiotics as treatment. Only 39.8% of the participants correctly selected Praziquantel as the drug of choice. The choice of drinking contaminated water as a mode of transmission of urinary schistosomiasis by close to 70% of those reported being hearing of it implies lack of knowledge related to transmission of the condition. Nearly a fifth of those aware of urinary schistosomiasis wrongly selected the use of insecticides as a preventive measure. The findings regarding treatment and prevention of urinary schistosomiasis contrast with those from Tanzania [17] where all the interviewed healthcare workers knew that Praziquantel was the drug of choice and recommended for treatment of schistosomiasis. The different study populations between the present study and the Tanzanian study would account for this observed difference.

The study was conducted in only the Volta Region of Ghana and the findings may be of limited generalizability. However, it may be also argued that the findings, especially those relating to FGS, will likely be similar in other regions since midwifery training institutions all over the country run the same curriculum accredited by the Ghana Tertiary Education Commission and the Nursing and Midwifery Council. The study is also limited by the use of 'straight-jacket' responses. A qualitative approach would have provided nuanced contexts surrounding the choice of wrong responses especially. Lastly, it would have been beneficial to conduct this study among the in-service cadre at this point of little to no knowledge of FGS but it was rather done among pre-service students. This hinders its generalizability to the in-service group of health workers. Nevertheless, the study findings raise concerns about the poor knowledge/ awareness of FGS among the study participants and constitute a reawakening call to increase efforts at making pre-service health workers know more about FGS.

## Conclusions

This study assessed the awareness of US and FGS among final year midwifery students in the Volta region of Ghana as well as the knowledge of their signs, symptoms, complications, treatment and prevention. Though awareness of urinary schistosomiasis was high, the study identified a significant number of the participants who have not heard of FGS and this reflected on their knowledge on the transmission, treatment and prevention of the condition. With the lack of comprehensive knowledge of US and low awareness of FGS, among the final year midwifery students, it is recommended that FGS be made part of the curriculum for the midwifery programme so that there could be increased awareness. Again, the Volta Regional Health Directorate must organize health campaigns through school activities, mass media, to help improve knowledge of urogenital schistosomiasis since the Volta Region has some areas that are endemic for schistosomiasis. Such increased awareness and/or knowledge is expected to impact positively on the future practice of this cadre of health workers at primary health care facilities.

## Supporting information

**S1 Checklist. *PLOS ONE* clinical studies checklist.**
(DOCX)

**S2 Checklist. Filled STROBE checklist.**
(DOCX)

**S1 Questionnaire. Questionnaire used for data collection.**
(DOCX)

**S1 Data. Data underlying the study findings.**
(XLSX)

## Acknowledgments

The authors are grateful to the participating students and heads of the three midwifery training institutions in the Volta region of Ghana.

## Author Contributions

**Conceptualization:** Wisdom Klutse Azanu, Joseph Osarfo, Verner Orish, Evans Kofi Agbeno, Emmanuel Senanu Komla Morhe.

**Data curation:** Gideon Appiah, Yvonne Sefadzi Godonu.

**Formal analysis:** Joseph Osarfo, Gideon Appiah, Gifty Dufie Ampofo, Emmanuel Senanu Komla Morhe, Margaret Gyapong.

**Funding acquisition:** Evans Kofi Agbeno.

**Investigation:** Wisdom Klutse Azanu, Joseph Osarfo, Gideon Appiah, Yvonne Sefadzi Godonu, Verner Orish, Michael Amoh.

**Methodology:** Wisdom Klutse Azanu, Joseph Osarfo, Yvonne Sefadzi Godonu, Gifty Dufie Ampofo, Verner Orish, Margaret Gyapong.

**Project administration:** Gideon Appiah, Yvonne Sefadzi Godonu.

**Supervision:** Wisdom Klutse Azanu, Joseph Osarfo, Michael Amoh.

**Validation:** Michael Amoh.

**Writing – original draft:** Wisdom Klutse Azanu, Joseph Osarfo.

**Writing – review & editing:** Wisdom Klutse Azanu, Joseph Osarfo, Gifty Dufie Ampofo, Verner Orish, Michael Amoh, Evans Kofi Agbeno, Emmanuel Senanu Komla Morhe, Margaret Gyapong.

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
