## [Decision Letter · Decision Letter 0]

26 Jan 2024

PONE-D-23-32022Knowledge of Female Genital and Urinary Schistosomiasis among final-year midwifery students in the Volta Region of Ghana.PLOS ONE

Dear Dr. Osarfo,

Thank you for submitting your manuscript to PLOS ONE. After careful consideration, we feel that it has merit but does not fully meet PLOS ONE’s publication criteria as it currently stands. Therefore, we invite you to submit a revised version of the manuscript that addresses the points raised during the review process.

We look forward to receiving your revised manuscript.

Kind regards,

Adetayo Olorunlana, Ph.D.

Academic Editor

PLOS ONE

A. You may seek permission from the original copyright holder of Figure 1 to publish the content specifically under the CC BY 4.0 license.  

B. If you are unable to obtain permission from the original copyright holder to publish these figures under the CC BY 4.0 license or if the copyright holder’s requirements are incompatible with the CC BY 4.0 license, please either i) remove the figure or ii) supply a replacement figure that complies with the CC BY 4.0 license. Please check copyright information on all replacement figures and update the figure caption with source information. If applicable, please specify in the figure caption text when a figure is similar but not identical to the original image and is therefore for illustrative purposes only.

Reviewers' comments:

Reviewer's Responses to Questions

**Comments to the Author**

1. Is the manuscript technically sound, and do the data support the conclusions?

Reviewer #1: Yes

Reviewer #2: Yes

2. Has the statistical analysis been performed appropriately and rigorously? 

Reviewer #1: Yes

Reviewer #2: Yes

3. Have the authors made all data underlying the findings in their manuscript fully available?

Reviewer #1: Yes

Reviewer #2: Yes

4. Is the manuscript presented in an intelligible fashion and written in standard English?

Reviewer #1: No

Reviewer #2: Yes

5. Review Comments to the Author

Reviewer #1: Knowledge of Female Genital and Urinary Schistosomiasis among final-year midwifery students in the Volta Region of Ghana.

Thank you for the opportunity to review this paper. This is a paper describing an assessment of knowledge level for urinary schistosomiasis and female genital schistosomiasis among final year midwifery students in Ghana. The authors should be commended for writing this paper on an important topic of neglected tropical diseases. However, there are some areas that will need to be revised to improve general quality of the paper. Below are general and specific comments for the authors’ consideration:

A. General comments

1) Remove unnecessary capitalization of first letters and abbreviations throughout the write up.

2) The title should reflect what is contained in the paper. While the title is about knowledge of urinary schistosomiasis and female genital schistosomiasis, in the main body, authors are mainly featuring FGS at expense of urinary schistosomiasis also appearing in the title. It is either the title to be rephrased or address the imbalance in the main body.

B. Title and abstract

2) The title of the paper please write female genital schistosomiasis in full. Since FGS is a subset of urinary schistosomiasis, authors should consider interposing these in the title to start with the broad urinary schistosomiasis then specifically with FGS.

3) As done with Background, Methodology, Findings and Conclusions should also be presented separately in the abstract.

C. Introduction

4) Since the paper is assessing knowledge of urinary schistosomiasis and FGS, authors should first separately introduce the context of urinary schistosomiasis before delving into FGS.

5) Lines 108-114 about urinary schistosomiasis should come earlier before FGS as pointed out in above comment.

6) Lines 115-135 authors should be justifying why it was necessary to carry out this study which is about the need for health workers (generally - not just midwives) having adequate knowledge about urinary schistosomiasis and FGS. Remove the ‘final year midwifery students’ bit from this part of the Introduction to Methods where study population is described.

D. Methodology

7) Study design, study site description and study population: Authors tend to merge several sub-sections into one making it difficult for readers to follow the write up. Consider breaking down the sub-heading into separate sub-sections i.e. Study design; Study area; Study population to make it easier for readers to follow the presentation.

E. Results

8) In presentation of the results, authors should start by presenting the results and refer to a Table or Figure towards the end not at the beginning as they have done with Table 1 in line 224.

9) Start by presenting the total number of participants before breaking down according to sites, age groups etc. i.e. “The study enrolled the total of XX participant of which ZZ (%) were from site A, YY (%) site B….”

10) Present numbers in digits not in words as done in line 227.

11) Ensure that correct results are presented under a correct heading. The first sub-heading is about “Background characteristics of participants” where authors should present characteristics of the participants such as age, sex, school etc. with help of a revised Table 1 (without the last two variables which need to be reported under own sub-headings). In line with the title of this paper authors should report for both urinary schistosomiasis and FGS. The next sub-heading is about “Knowledge of Female Genital Schistosomiasis” should come earlier where all results of participants’ knowledge about FGS (including one (last) variable in Table 1 should be presented. Similarly, authors should consider preceding this section with a sub-heading on “Knowledge of Urinary Schistosomiasis” on line 277 to appear earlier before FGS as indicated in 2 and 4 above. Tables 2-5 and their respective narratives should be realigned with appropriate sub-heading.

F. Discussions

12) The discussion should be re-organized to include findings for both urinary schistosomiasis and FGS (making sure that urinary schistosomiasis coming before FGS.

Reviewer #2: Manuscript Number: PONE-D-23-32022

Overall Comments

The paper is of interest to all health care practitioners more specifically those working in areas of sexual and reproductive health, and I believe in all parts of SSA where schistosomiasis is endemic. Lines 55-64 speak to the primary purpose which was to assess final year midwifery students in the Volta Region of Ghana an area endemic for urinary schistosomiasis for knowledge of causes, transmission, presentation, treatment and prevention of FSG. Authors highlight key issues: missed diagnosis due to limited knowledge etc. During training, particularly in the clinical area aren't women screened schistosomiasis for family planning, particularly for IUCD and for prenatal care to rule out both STI and schistosomiasis given that it is endemic in that region; when the activities on Neglected Tropical Diseases is aiming for a global elimination?

Line 472 Conclusions

while I agree with recommendations made, one cannot help but wonder what the Ghana national policy on prevention and control says about FSG? No reference was made to. Shouldn't all health training institutions curricula for doctors, nurses, lab technicians/scientists for example include schistosomiasis? given the limited knowledge regarding causes, diagnostic, treatment and prevention and control?

6. PLOS authors have the option to publish the peer review history of their article (what does this mean?). If published, this will include your full peer review and any attached files.

Reviewer #1: **Yes: **Peter Makaula

Reviewer #2: **Yes: **Nthabiseng Phaladze

---

## [Author Response · Author response to Decision Letter 0]

5 Feb 2024

RESPONSES TO REVIEWERS’ COMMENTS AND JOURNAL REQUIREMENTS

JOURNAL REQUIREMENTS

1. Copyright issues with Figure 1.

Response: On the matter of the potential copyrights relating to Figure 1, the authors have chosen to remove the said image/figure. We have no means of identifying the original copyright holders and therefore will not take this path. Hence, Figure 1 is removed from the submission and all references to it in the manuscript have also been taken out.

RESPONSES TO COMMENTS FROM REVIEWER #1

1. Remove unnecessary capitalization of first letters and abbreviations throughout the write up.

Response: All unnecessary capitalization of first letters and abbreviations have been removed as suggested. Only those deemed necessary have been left.

2. The title should reflect what is contained in the paper. While the title is about knowledge of urinary schistosomiasis and female genital schistosomiasis, in the main body, authors are mainly featuring FGS at expense of urinary schistosomiasis also appearing in the title. It is either the title to be rephrased or address the imbalance in the main body.

Response: This has been addressed. We have reported on knowledge levels regarding urinary schistosomiasis as well and provided appropriate citations. Please see the second paragraph of the ‘Introduction’ which reads as follows;

3. The title of the paper please write female genital schistosomiasis in full. Since FGS is a subset of urinary schistosomiasis, authors should consider interposing these in the title to start with the broad urinary schistosomiasis then specifically with FGS.

Response: Female genital schistosomiasis has been written out fully and the title maintained.

4. As done with Background, Methodology, Findings and Conclusions should also be presented separately in the abstract.

Response: This has been corrected and we are grateful to the reviewer for drawing our attention to this anomaly. This came about because the manuscript was originally formatted for and submitted to a sister PLOS journal (PLOS NTDs). They advised it was more suitable for PLOS ONE and subsequently forwarded it. 

5. Since the paper is assessing knowledge of urinary schistosomiasis and FGS, authors should first separately introduce the context of urinary schistosomiasis before delving into FGS.

Response: We agree with the reviewer on this and have presented a slight variation in the revised manuscript. In the opening paragraph of the ‘Introduction’, we have established the link between urinary schistosomiasis (US) and female genital schistosomiasis (FGS). We chose to describe the clinical implications of FGS first to emphasize our focus on it. Subsequently, we highlighted knowledge levels/ local prevalence of US and compared it to that of FGS. Eventually, we went into the problem statement, study objective and study significance where it has been emphasized that we added on US for context and to enable comparison of knowledge base of these two conditions. Thank you.

6. Lines 108-114 about urinary schistosomiasis should come earlier before FGS as pointed out in above comment.

Response: This has been done (though with a slightly varying approach).

7. Lines 115-135 authors should be justifying why it was necessary to carry out this study which is about the need for health workers (generally - not just midwives) having adequate knowledge about urinary schistosomiasis and FGS. Remove the ‘final year midwifery students’ bit from this part of the Introduction to Methods where study population is described.

Response: The last sentence in the last-but-two paragraph of the ‘Introduction’ describes fall- outs from a lack of adequate knowledge of FGS among health care workers and justifies the need to assess this knowledge among health care workers including those still in training (as outlined in the new opening sentence of the penultimate paragraph……” It is therefore important to assess the knowledge of health care providers, including those still in training, regarding FGS.”). 

The authors humbly disagree with the reviewer concerning moving ‘the final year midwifery students’ bit from the Introduction to the Methods section. The authors are of the opinion that the specific study population and the justification for that study population need to be clarified early in the context of defining the problem statement and the study significance.

8. Study design, study site description and study population: Authors tend to merge several sub-sections into one making it difficult for readers to follow the write up. Consider breaking down the sub-heading into separate sub-sections i.e. Study design; Study area; Study population to make it easier for readers to follow the presentation.

Response: As suggested and to improve clarity, ‘Study design and population’ has now been made a separate sub-section while ‘Study site description’ is another sub-section. Please see the version with track changes and the clean version as well.

9. In presentation of the results, authors should start by presenting the results and refer to a Table or Figure towards the end not at the beginning as they have done with Table 1 in line 224.

Response: The recommendation has been accepted and revision done accordingly. Presentation of participants’ background characteristics (which have been so defined to include whether they have ever heard of urinary schistosomiasis and female genital schistosomiasis) now reads as follows;

“All planned 193 final-year students took part in the survey. Hohoe and Keta NMTC had similar numbers of participants in the study; 71 (36.8%) and 73 (37.8%). Close to half (48.7%) were in the age group 23-27 years while 31 (16.1%) were in the youngest age group, 18-22 years One hundred and twenty-three (64.1%) participants indicated they had heard about urinary schistosomiasis. Comparatively, just about a quarter (23.3%) reported ever hearing of female genital schistosomiasis (see Table 1).”

10. Start by presenting the total number of participants before breaking down according to sites, age groups etc. i.e. “The study enrolled the total of XX participant of which ZZ (%) were from site A, YY (%) site B….”

Response: The recommendation is accepted and revision has been done accordingly. The relevant section now reads as follows;

“All planned 193 final-year students took part in the survey. Hohoe and Keta NMTC had similar numbers of participants in the study; 71 (36.8%) and 73 (37.8%). Close to half (48.7%) were in the age group 23-27 years while 31 (16.1%) were in the youngest age group, 18-22 years One hundred and twenty-three (64.1%) participants indicated they had heard about urinary schistosomiasis. Comparatively, just about a quarter (23.3%) reported ever hearing of female genital schistosomiasis (see Table 1).”

11. Present numbers in digits not in words as done in line 227.

Response: We thank the reviewer for his comments. In the portion under reference, the number begins a sentence. The convention is to have it in words under such circumstances. The authors therefore prefer to maintain the status quo.

12. Ensure that correct results are presented under a correct heading. The first sub-heading is about “Background characteristics of participants” where authors should present characteristics of the participants such as age, sex, school etc. with help of a revised Table 1 (without the last two variables which need to be reported under own sub-headings). In line with the title of this paper authors should report for both urinary schistosomiasis and FGS. 

Response: We are grateful to the reviewer for his comments. In the particular case of Table 1, the last two variables “ever heard of urinary schistosomiasis” and “ever heard of female genital schistosomiasis” have been defined to be part of ‘Background Characteristics’ (which is broader than just demographic characteristics) in the Methods section. The authors opine that ‘ever heard’ does not exactly equate to ‘knowledge’ and thus the two should not be together.

Under Methods, the Study procedure and data collection sub-section has been revised to read;

“……….The questionnaire was divided into three main parts. The first part collected data on the background characteristics of participants which included age, institution attended and whether they had ever heard of urinary and female genital schistosomiasis……….”

On account of this, the authors prefer to maintain Table 1 in its current form.

13. The next sub-heading is about “Knowledge of Female Genital Schistosomiasis” should come earlier where all results of participants’ knowledge about FGS (including one (last) variable in Table 1 should be presented. Similarly, authors should consider preceding this section with a sub-heading on “Knowledge of Urinary Schistosomiasis” on line 277 to appear earlier before FGS as indicated in 2 and 4 above. Tables 2-5 and their respective narratives should be realigned with appropriate sub-heading.

Response: This is acknowledged. The suggestions have been adopted and presented in a slightly different way to improve clarity. A new sub-section “Participants’ source of information about urinary and female genital schistosomiasis” has been introduced and the old Table 2 is now Table 3. “Knowledge of urinary schistosomiasis” now precedes that of female genital schistosomiasis and the tables / table numbers have been realigned with their sub-headings appropriately.

14. The discussion should be re-organized to include findings for both urinary schistosomiasis and FGS (making sure that urinary schistosomiasis coming before FGS).

Response: We have included findings on urinary schistosomiasis in the opening paragraph of the Discussion. It now reads as follows;

“………Among those who reported hearing about urinary schistosomiasis before, about 93% correctly chose ‘blood in urine’ as a symptom while 68% wrongly selected inguinal hernia as a symptom or indicated they did not know if it was or not. Nearly 70% of these specific participants indicated that urinary schistosomiasis was transmitted by drinking contaminated water.”

Findings on both urinary schistosomiasis and female genital schistosomiasis have been discussed vis-à-vis each other to better relate comparisons in practically all of the paragraphs of the ‘Discussion’. 

RESPONSES TO COMMENTS FROM REVIEWER #2

1. The paper is of interest to all health care practitioners more specifically those working in areas of sexual and reproductive health, and I believe in all parts of SSA where schistosomiasis is endemic. Lines 55-64 speak to the primary purpose which was to assess final year midwifery students in the Volta Region of Ghana an area endemic for urinary schistosomiasis for knowledge of causes, transmission, presentation, treatment and prevention of FSG. Authors highlight key issues: missed diagnosis due to limited knowledge etc. During training, particularly in the clinical area aren't women screened schistosomiasis for family planning, particularly for IUCD and for prenatal care to rule out both STI and schistosomiasis given that it is endemic in that region; when the activities on Neglected Tropical Diseases is aiming for a global elimination?

Response: The authors are grateful to the reviewer for the kind comments. 

We do screen for STIs as part of work-ups for uptake of IUCD and for prenatal care as mentioned. Unfortunately, schistosomiasis has never been on the agenda in this respect. As mentioned in the ‘Background’ of the manuscript, some vaginal discharges may be caused by FGS but because our minds were not focused to it, the tendency has been to treat with antibiotics. In some circumstances, one may see some improvement but this is soon followed by a recurrence. In others, there is no improvement at all since not all vaginal discharge is caused by bacteria and patients keep on moving from one health facility to the next seeking for solutions.

In summary, FGS has not been the radar at all. In the routine clinical care and family planning services in Ghana, women are not routinely screened for schistosomiasis or FGS unless they are symptomatic. In many endemic areas like Ghana, there is rather MDA from time to time to help control/prevent urinary schistosomiasis/FGS. Prenatal care includes routine urine examination which can pick up schistosomiasis but not FGS. Besides, we don’t do routine colposcopy for prenatal services. Even in performing colposcopy for other reasons, most health care providers lack the awareness and/or knowledge to pick up FGS

2. while I agree with recommendations made, one cannot help but wonder what the Ghana national policy on prevention and control says about FSG? No reference was made to. Shouldn't all health training institutions curricula for doctors, nurses, lab technicians/scientists for example include schistosomiasis? given the limited knowledge regarding causes, diagnostic, treatment and prevention and control?

Response: Ghana has a programme for NTD control that oversees schistosomiasis, lymphatic filariasis, onchocerciasis, yaws and others. However, FGS has never been in the picture. Schistosomiasis, as a topic, is in the curriculum of the health training institutions but there is really no mention of FGS. Thus, there has never been any established education on FGS. Until recent awareness creation by the FGS FAST package by a partnership of global and local actors to fund and support awareness creation about FGS, most of the faculty in health training institutions (including very senior gynaecologists as we realized in a recent West African conference in Lome, Togo) were not aware of FGS. It is expected that the study findings will contribute to making a case for incorporating FGS into the curriculum of health training institutions.

---

## [Decision Letter · Decision Letter 1]

11 Mar 2024

PONE-D-23-32022R1Knowledge of female genital schistosomiasis and urinary schistosomiasis among final-year midwifery students in the Volta Region of Ghana.PLOS ONE

Dear Dr. Osarfo,

Thank you for submitting your manuscript to PLOS ONE. After careful consideration, we feel that it has merit but does not fully meet PLOS ONE’s publication criteria as it currently stands. Therefore, we invite you to submit a revised version of the manuscript that addresses the points raised during the review process.

We look forward to receiving your revised manuscript.

Kind regards,

Adetayo Olorunlana, Ph.D.

Academic Editor

PLOS ONE

Reviewers' comments:

Reviewer's Responses to Questions

**Comments to the Author**

1. If the authors have adequately addressed your comments raised in a previous round of review and you feel that this manuscript is now acceptable for publication, you may indicate that here to bypass the “Comments to the Author” section, enter your conflict of interest statement in the “Confidential to Editor” section, and submit your "Accept" recommendation.

Reviewer #1: (No Response)

2. Is the manuscript technically sound, and do the data support the conclusions?

Reviewer #1: Yes

3. Has the statistical analysis been performed appropriately and rigorously? 

Reviewer #1: Yes

4. Have the authors made all data underlying the findings in their manuscript fully available?

Reviewer #1: Yes

5. Is the manuscript presented in an intelligible fashion and written in standard English?

Reviewer #1: Yes

6. Review Comments to the Author

Reviewer #1: Knowledge of female genital schistosomiasis and urinary schistosomiasis among final-year midwifery students in the Volta Region of Ghana.

Thank you for the opportunity to again review the revised paper following comments made by reviewers. The authors should be commended for addressing most of concerns that were raised which has now improved the manuscript. However, there are still some areas that will need to be revised. Below are some outstanding and emerging comments for the authors’ consideration:

A. Introduction

1) Lines 110-119 are repeating what has already been reported in lines 87-95 citing the same Nigerian study [13].

2) Line 126-127 – The sentence “It is therefore important to assess the knowledge of health care providers, including those still in training, regarding FGS.” should be moved to the next paragraph at the beginning of line 138.

3) Lines 139-141 – authors write “The Volta region was chosen because, historically, communities along the Volta Lake have been known to be endemic for urinary schistosomiasis [15, 27].” Since the purpose of this study was to assess the acquired knowledge/training about US and FGS among midwifery students it is not relevant whether the chosen area was to be an endemic for schistosomiasis or not.

B. Methodology

4) [A previous comment]: Authors tend to merge several sub-sections into one making it difficult for readers to follow the write up. Consider breaking down the sub-heading into separate sub-sections i.e. Study design; Study area; Study population to make it easier for readers to follow the presentation. This comment remains unresolved. For example, in the revised manuscript, authors have chosen to merge ‘study design with population’ yet the content therein have elements of time and place of the study not related to the subheading. There are so many instances of repetitions within the subheadings, the authors are therefore invited to critically re-examine each sub-heading and the contents to ensure that they are not repeated, easy and logical for a reader to follow.

5) Study site description – authors are presenting data from OPD or DHMIS2 probably to justify why the areas were selected for the study. This again raises the question in 3 above, what have these statistics to do with choice of the midwifery students who were the study population as opposed to community members whose data was recorded at OPD/DHMIS2?

6) Sample size estimation should be preceded by a ‘study population description’ sub-section, either separately or merged.

7) Study procedures and data collection – line 184 authors write “All final-year midwifery students in the Volta region were included in the study.” This sentence is contradictory with what is immediately written up to line 197.

B. Discussions

8) Since the authors indicated that in Ghana only US unlike FGS is part of curricula in medical training institutions. The authors need to explore and discuss the obtained result in the present study that indicated schools/workshops as the main source of information about US (54.9%) and FGS (59%) for many of the midwifery students that were interviewed. Were these schools/workshops where they got information at the same training colleges they were enrolled at the time of the study or different ones since these are not yet qualified midwives?

9) In lines 347-350 authors write “Participants from Keta NMTC had the highest proportion of participants who had heard about urinary schistosomiasis followed by UHAS and Hohoe NMTC respectively. This could be because Keta has a lot more water bodies than the other places and possibly a higher burden of urinary schistosomiasis.” Can authors explain the connection between the environmental setting of the area with the respondent midwifery students’ acquired knowledge/training in relation to comments 3 and 5 above?

10) Another limitation of the study is that it was conducted among pre-service students, hindering its generalizability to the in-service group of health worker cadres who acquire more exposure and experience while interacting with community members than the students who are mostly attending lessons in their colleges.

C. Conclusions

11) The first sentence should qualify “[…] among final year midwifery students in the Volta Region, Ghana”.

7. PLOS authors have the option to publish the peer review history of their article (what does this mean?). If published, this will include your full peer review and any attached files.

Reviewer #1: **Yes: **Peter Makaula

---

## [Author Response · Author response to Decision Letter 1]

24 Mar 2024

RESPONSES TO SECOND ROUND OF COMMENTS FROM REVIEWER #1

1) Lines 110-119 are repeating what has already been reported in lines 87-95 citing the same Nigerian study [13].

Response: The authors are thankful to the reviewer for this comment. This specific comment is from the last review and was addressed. Thus, lines 110-119 are not a repetition of lines 87-95. The Nigerian study [13] has been used in both portions to represent differing contexts.

2) Line 126-127 – The sentence “It is therefore important to assess the knowledge of health care providers, including those still in training, regarding FGS.” should be moved to the next paragraph at the beginning of line 138.

Response: This has been done as suggested.

3) Lines 139-141 – authors write “The Volta region was chosen because, historically, communities along the Volta Lake have been known to be endemic for urinary schistosomiasis [15, 27].” Since the purpose of this study was to assess the acquired knowledge/training about US and FGS among midwifery students it is not relevant whether the chosen area was to be an endemic for schistosomiasis or not.

Response: We thank the reviewer for his insightful comments. In this particular instance, we reckoned that since US is endemic in the communities referred to, the trainee midwives would, at least, have a minimum acceptable level of knowledge of US and FGS. The trainee midwives are, at the barest minimum, part of the population in these areas and this has been alluded to in the Discussion section originally. The section in question has been revised to include a second reason for working in Volta Region and now reads as;

“The Volta region was chosen because communities along the Volta Lake have been known to be endemic for urinary schistosomiasis [15, 27] and also for ease of accessibility to the students as all but one of the investigators are based in the region.”

4) [A previous comment]: Authors tend to merge several sub-sections into one making it difficult for readers to follow the write up. Consider breaking down the sub-heading into separate sub-sections i.e. Study design; Study area; Study population to make it easier for readers to follow the presentation. This comment remains unresolved. For example, in the revised manuscript, authors have chosen to merge ‘study design with population’ yet the content therein have elements of time and place of the study not related to the subheading. There are so many instances of repetitions within the subheadings, the authors are therefore invited to critically re-examine each sub-heading and the contents to ensure that they are not repeated, easy and logical for a reader to follow.

Response: This has been done as suggested. The ‘Study population’ now stands alone just before sample size estimation. The period of data collection has been moved to ‘Study procedures and data collection’. The authors acknowledge and appreciate the differences in presentation style but we would like to maintain the hint of ‘place of study’ under the ‘Study design’.

5) Study site description – authors are presenting data from OPD or DHMIS2 probably to justify why the areas were selected for the study. This again raises the question in 3 above, what have these statistics to do with choice of the midwifery students who were the study population as opposed to community members whose data was recorded at OPD/DHMIS2?

Response: The authors are grateful to the reviewer for the opportunity to explain our rationale. First, we did not exactly ‘select’ the study areas. They are included by default because those areas are where the three midwifery training institutions in the Volta Region are located. This was indicated earlier in the Methods section. The health facility data provided give context of the burden of urinary schistosomiasis (US) in the various areas as part of the study site description. To our knowledge and as much as possible, study site description ought to be done with a focus on some aspects of the subject matter in question. Since ours is the first evaluation among midwifery students in our country, we reckoned we could at least minimally describe the amount of documented cases of US. The said data did not play any role at all in the choice of the study population.

6) Sample size estimation should be preceded by a ‘study population description’ sub-section, either separately or merged

Response: This has been done. ‘Study population’ stands alone just before the sub-section on sample size estimation.

7) Study procedures and data collection – line 184 authors write “All final-year midwifery students in the Volta region were included in the study.” This sentence is contradictory with what is immediately written up to line 197.

Response: We thank the reviewer for picking up this oversight. The sentence in question is now revised to read as;

“All final-year midwifery students in the Volta region were eligible for inclusion in the study.”

8) Since the authors indicated that in Ghana only US unlike FGS is part of curricula in medical training institutions. The authors need to explore and discuss the obtained result in the present study that indicated schools/workshops as the main source of information about US (54.9%) and FGS (59%) for many of the midwifery students that were interviewed. Were these schools/workshops where they got information at the same training colleges they were enrolled at the time of the study or different ones since these are not yet qualified midwives?

Response: This study finding was originally discussed over lines 336-349. We have further revised and strengthened the Discussion on these observations. Specifically, lines 343-352 now read as;

“……In comparison, FGS is so far not part of the curriculum in their training. It is probable then that the knowledge of FGS was more from extracurricular workshops organized for these midwives. For instance, it is possible that their tutors may have been involved in training workshops about schistosomiasis and FGS organized by the Volta River Authority (VRA) and may have casually handed down such information during lectures on related topics such as anaemia in pregnancy. In addition to the Neglected Tropical Diseases Control Programme in Ghana, the VRA is also involved in schistosomiasis control along the Volta Lake. Nevertheless, here in Ghana, the named sources may have to be taken with caution since FGS is still a relatively new ‘discovery’ that is now being brought to the attention of even clinicians.”

9) In lines 347-350 authors write “Participants from Keta NMTC had the highest proportion of participants who had heard about urinary schistosomiasis followed by UHAS and Hohoe NMTC respectively. This could be because Keta has a lot more water bodies than the other places and possibly a higher burden of urinary schistosomiasis.” Can authors explain the connection between the environmental setting of the area with the respondent midwifery students’ acquired knowledge/training in relation to comments 3 and 5 above?

Response: In relation to comment 5 specifically, we sought to describe the study area with respect to documented occurrences of US cases. The argument (if the authors may be allowed to call it so) with respect to comments 3 and 9 has been that the midwifery students, being part of the general population and also being part of the hospital setting at different times in their training, may be presumed have some minimum level of awareness / knowledge of US in areas with relatively more appreciable burdens of US. The particular example of Keta is made where the students live within communities rather than secluded hostels and may be more aware of mass drug administration efforts for US.

The section (lines 358-369) in question is now revised to read as follows;

“Keta NMTC had the highest proportion of participants who had heard about urinary schistosomiasis followed by UHAS and Hohoe NMTC respectively. This could be because Keta has a lot more water bodies than the other places and possibly a higher burden of urinary schistosomiasis which may underpin relatively more awareness among these students as their training also involves the hospital and community setting. The mass drug administration for prevention of Schistosoma infection in that area may also have created more awareness. Also in Keta NMTC, most of the students live within the community rather than in secluded hostels and so may be more aware of or involved in community activities pertaining to urinary schistosomiasis than in UHAS and Hohoe where the students live in hostels on the schools’ premises. This relation between the school attended and whether the respondents had ever heard of urinary schistosomiasis was not observed in the case of FGS. This may arise from the general deficiency in knowledge of FGS.”

10) Another limitation of the study is that it was conducted among pre-service students, hindering its generalizability to the in-service group of health worker cadres who acquire more exposure and experience while interacting with community members than the students who are mostly attending lessons in their colleges

Response: The suggested limitation has been included.

11) The first sentence should qualify “[…] among final year midwifery students in the Volta Region, Ghana”.

Response: This was an oversight and we thank the reviewer for detecting it. The suggested qualification has been inserted appropriately and the sentence now reads as;

“This study assessed the awareness of US and FGS among final year midwifery students in the Volta region of Ghana as well as the knowledge of their signs, symptoms, complications, treatment and prevention.”

---

## [Decision Letter · Decision Letter 2]

8 Apr 2024

Knowledge of female genital schistosomiasis and urinary schistosomiasis among final-year midwifery students in the Volta Region of Ghana.

PONE-D-23-32022R2

Dear Dr. Osarfo,

We’re pleased to inform you that your manuscript has been judged scientifically suitable for publication and will be formally accepted for publication once it meets all outstanding technical requirements.

Kind regards,

Adetayo Olorunlana, Ph.D.

Academic Editor

PLOS ONE

Additional Editor Comments (optional):

Reviewers' comments:

Reviewer's Responses to Questions

**Comments to the Author**

1. If the authors have adequately addressed your comments raised in a previous round of review and you feel that this manuscript is now acceptable for publication, you may indicate that here to bypass the “Comments to the Author” section, enter your conflict of interest statement in the “Confidential to Editor” section, and submit your "Accept" recommendation.

Reviewer #1: All comments have been addressed

2. Is the manuscript technically sound, and do the data support the conclusions?

Reviewer #1: Yes

3. Has the statistical analysis been performed appropriately and rigorously? 

Reviewer #1: Yes

4. Have the authors made all data underlying the findings in their manuscript fully available?

Reviewer #1: Yes

5. Is the manuscript presented in an intelligible fashion and written in standard English?

Reviewer #1: Yes

6. Review Comments to the Author

Reviewer #1: Thank you for the opportunity to re-review the revised manuscript following comments I made. The authors have now addressed all the concerns that were raised which has led to a significant improvement of the manuscript. I have no further comments for the authors.

7. PLOS authors have the option to publish the peer review history of their article (what does this mean?). If published, this will include your full peer review and any attached files.

Reviewer #1: **Yes: **Peter Makaula
